# Profile of individuals served and presumed coverage of Psychosocial Care Centers (CAPS) in Brazil: A study of the period 2013–2019

**Bruna Paiva do Carmo Mercedes**[1]*, **Everton Nunes da Silva**[2], **Rodrigo Luiz Carregaro**[3], **Adriana Inocenti Miasso**[1]

1 Ribeirão Preto Nursing College, University of São Paulo, Ribeirão Preto, São Paulo, Brazil, 2 School of Collective Health, Ceilândia College, University of Brasília, Brasília, Brazil, 3 School of Physiotherapy, Ceilândia College, University of Brasília, Brasília, Brazil

* brunamercedes86@gmail.com

## Abstract

### Objective

To describe the profile of individuals with depression who received assistance at CAPS in Brazil between 2013 and 2019, focusing on their clinical and demographic characteristics, and to calculate the estimated coverage of CAPS across the national territory and its constituent federal units.

### Methods

Descriptive, ecological, time-series study with secondary data from national databases, referring to care provided at CAPS in the country for adults aged 18 years or over-diagnosed with depression (F32-32.9 and F33-F33.9). The estimated coverage of CAPS was calculated for 2013 and 2019 using registered and active services.

### Results

There was a 107% increase in the number of patients with depression receiving treatment at CAPS between 2013 and 2019. Women accounted for 77% of the patients, with the majority falling within the age range of 41 to 61years (49%). The predominant racial demographics were white (38%) and brown (34%). The diagnosis of depressive episodes was prevalent among 65% of the patients, and individual care was administrated to 75% of them. The presumed CAPS coverage was 71% in 2013 and increased to 87% in 2019 nationwide, although significant discrepancies were observed among different states.

### Conclusion

Progress was evident in the implementation of CAPS across Brazil during the period spanning 2013 to 2019. Nonetheless, disparities persist among the federative units, and there remains an underutilization of group and family care within CAPS services.

**Data Availability Statement:** All relevant data are within the manuscript and its Supporting information files.

**Funding:** The author(s) received no specific funding for this work.

**Competing interests:** The authors declared that there are no conflict interests.

## Introduction

Currently, depression stands as the foremost mood disorder prevalent in society. This condition afflicts over 280 million individuals globally, with an estimated 5% of adults affected by depression [1]. The World Health Organization (WHO) identifies depression as a major contributor to disability, exerting significant impact on the global burden of disease [2]. Consequently, it imposes substantial costs on healthcare and social security systems, alongside precipitating premature mortality through suicide [3]. Depressive disorders predominantly impact the young and economically active population [4]. Throughout the Americas, depression ranks as one of the most debilitating illnesses, contributing to 7.8% of the total disability burden in the population [4]. In South America, the ranking of disability attributed to depression includes five countries: Paraguay (9.4%), Brazil (9.3%), Peru (8.6%), Ecuador (8.3%), and Colombia (8.2%) [4].

Depression can be classified into depressive episodes or recurrent depressive disorders, commonly referred to as major depressive disorder. The distinction between these two types lies in the frequency of episodes and the severity of symptoms. Depressive disorders are considered chronic rather than episodic when an individual experiences at least two previous depressive episodes. Both types are characterized by a persistent low mood or loss of ability to experience pleasure (anhedonia), along with other cognitive, behavioral, and neurovegetative symptoms [2, 4, 5] that impact the individual's functional capacity [6].

In Brazil, the prevalence of depression among individuals aged 18 years or older was 7.6% in 2013, as reported by one study [7]. By 2019 [8], this figure had risen to 10.2%, reflecting a notable increase of 34.2% over the six-year period. The escalation of this disorder within the population is alarming, underscoring the urgency for early detection by health professionals, particularly within primary health care (PHC) settings. In instances where PHC is unable to provide sufficient therapeutic support to the individual, referral to specialized care is imperative [9].

The inception of the Brazilian Psychiatric Reform dates back to the 1970s [10]; however, its transformative impact on the healthcare model was notably realized following the enactment of the National Policy on Mental Health, Alcohol, and Other Drugs in 2001. A pivotal legislative measure in this trajectory was Law 10.216/01, which mandated the transition from a psychiatric hospital-centric approach to community-based mental health services [11]. Subsequently, in 2002, the establishment of Psychosocial Care Centers (CAPSs) emerged as specialized mental health facilities, serving as a cornerstone in the reorganization of mental health services nationwide [12]. These CAPSs extend daily support to the designated population within their jurisdiction, offering clinical monitoring and psychosocial rehabilitation for individuals experiencing mental distress. Operated by multidisciplinary teams, CAPSs deliver personalized care through individual and group sessions, therapeutic workshops, home visits, family interventions, and community engagements, among other modalities [13].

In 2022, the Ministry of Health introduced the Depression Care Line for Adults, delineating guidelines for healthcare professionals operating within this domain [9]. Primary Health Care (PHC) assumes a pivotal role as the coordinating network and service facilitator. Each care juncture is accompanied by a structured referral pathway, initial intervention, and therapeutic planning, tailored to meet the specific needs and preferences of each patient. The care continuum outlined in this framework encompasses Primary Health Units, Specialized Care (CAPS), Emergency Care Units, Mobile Assistance Service (SAMU), and Hospital Units [9].

The increase of CAPS facilities across Brazil has been notable. In 2006, the quantity was 739, by 2014 it rose to 2,209 and further, by 2018 [3, 14], it reached 2,306 facilities, albeit punctuated by setbacks observed in the National Mental Health Policy (PNSM) post-2016. These

setbacks were characterized by a surge in investments in institution-based services, such as hospital admissions and therapeutic communities [14, 15], thereby impeding the process of social reintegration and exacerbating the stigma surrounding individuals with mental health conditions.

The National Mental Health Policy (PNSM) advocates for community-based care provided by CAPSs, aiming to mitigate inequalities and extend treatment to diverse populations. It emphasizes comprehensive care and ensures that individuals receive coordinated and continuous interventions across all levels of care. These principles align with those of the Unified Health System (SUS) and should guide care planning efforts. However, a significant challenge lies in understanding the demographics and characteristics of the served clientele to tailor service actions and promote equitable care. The objective goes beyond merely expanding service coverage; it involves coordinating Care networks and pathways regionally in a country with vast dimensions. Given the economic diversity and varying population sizes of municipalities, there is a pressing need to ensure qualified and effective assistance [16].

Against this backdrop, it becomes imperative to scrutinize the evolution of depression care within CAPSs in Brazil. Such insights are invaluable for policymakers in refining the management of depression within the SUS framework. Consequently, this study endeavors to describe the demographic and clinical profiles of individuals seeking depression care at CAPSs from 2013 to 2019, while also assessing the presumed coverage of CAPS facilities across different Brazilian states to identify potential regional disparities in healthcare infrastructure provision.

## Method

### Study design

The study conducted was a descriptive, ecological, and time series analysis utilizing secondary data obtained from national databases of the public health service. The data spanned from 2013 to 2019 [17].

### Context of the study

The context of the study pertains to the care delivered within the framework of Community Psychosocial Care Centers (CAPSs) across the national territory. These centers are equipped with multidisciplinary teams dedicated to the treatment of individuals with severe and persistent mental disorders [18]. Various modalities of CAPSs exist, distinguished by: i) the population size of the health region where the CAPS is located; ii) the demographics served, encompassing children and adolescents under 18 years old, adults aged 18 years or older, and individuals struggling with substance abuse; iii) operational hours, whether part-time (daytime) or full-time (24 hours); and iv) the array of services offered, including brief consultations or crisis interventions, group, individual, and family therapeutic activities, as well as overnight care or observation periods [19, 20]. The establishment of CAPS is the responsibility of municipalities or health regions, which receive dedicated funding from the Ministry of Health for this purpose. Given the tripartite nature of funding in the Unified Health System (SUS), these allocations may be supplemented by resources from the states and municipalities where CAPSs serve the enrolled population.

### Study population

The study population was based on the following inclusion criteria:

1. individuals with a medical diagnosis of depression according to the ICD-10, with codes F32-F32.9 (depressive episodes) and F33-F33.9 (recurrent depressive disorders). It was

considered only ICD-10 codes based on the principal diagnosis, which occasioned the need for the outpatient care.

2. individuals aged 18 years or older. We opted for this cut-off because CAPSs are essentially design to adult population. Few CAPSs are targeted to provide mental care to children and adolescents, focusing on their suffering and cognitive development.

3. individuals who used CAPSs at any point between 2013 and 2019.

   The exclusion criteria are:

1. individuals who received care at CAPSs but were not registered in the administrative records of the Ministry of Health for any reason. This could include instances where necessary information, such as the ICD-10 code or patient's ID, was not provided to the Ministry of Health.

2. individuals who utilized services other than those provided by CAPSs.

## Data collection and study variables

The data were sourced from a national database, specifically collected from the Outpatient Information System of the Unified Health System (SIA/SUS) via the Records of Outpatient Health Actions (RAAS), specifically focusing on RAAS—psychosocial. RAAS was utilized to capture specialized mental health care provided by Community Psychosocial Care Centers (CAPS) [21] from 2013 to 2019. The selection of this time frame was based on the commencement of registering consultations at CAPS in this application in 2013, as well as being the last year before the onset of the COVID-19 pandemic in 2019. It was reasoned that data entered into the system from 2020 onwards might not accurately reflect the reality of care due to significant changes in healthcare operations prompted by the pandemic. During this period, health authorities implemented various guidelines to combat the pandemic, including reallocating healthcare personnel and resources to COVID-19 efforts, thereby reducing the provision of non-COVID-19-related services. Additionally, the population expressed concerns about the pandemic, leading to delays or avoidance of medical care as a precautionary measure against the risk of contracting SARS-CoV-2 [21].

In the RAAS system, each procedure performed is assigned a specific code representing the various forms of care available at CAPS [22]: i) individual patient care in the CAPS (030108020–8): This involves tailored therapeutic modalities based on the individual's needs; ii) Patient group care at CAPS (030108021–6): Actions conducted collectively, utilizing group dynamics to address various purposes; iii) body practices (030108027–5): Activities focusing on enhancing body perception, self-image, psychomotor coordination, and somatic and postural aspects; iv) daytime reception of patients in CAPS (030108019–4): Daytime hospitality actions aimed at promoting the user's distancing from stressful situations; v) family care at CAPS (030108022–4): Actions directed towards the individual or collective care of family members and addressing their needs, vi) home visit (030108024–0): External activity conducted and supervised by a professional nurse with pre-established objectives; vii) psychosocial rehabilitation actions (030108024–0): Strategic actions promoting collaboration with other care points in health, education, justice, social assistance, human rights, and other networks; viii) overnight reception (030108002–0): Providing night-time hospitality for users already under the follow-up service; ix) reception in the third shift (030108002–0): Assistance provided between 6 pm and 9 pm; x) expressive and communicative practices (030108028–3): Strategies or activities aimed at expanding users' communicative and expressive abilities; xi) attention to

crises (030108028–3): Actions developed to manage crises effectively; xii) promotion of contractually (030108028–3): Monitoring users in real-life contexts; xiii) intra and intersectoral networking actions (030108025–9): Measures aimed at ensuring the subject's participation in formulating public and private policies regarding health promotion and prevention; xiv) strengthening the protagonism of CAPS users and their families (030108026–7): Activities encouraging the participation of users and their families in service management processes and the care network; xv) matrix support for Primary Care Teams (030108030–5): Facilitating shared construction between two or more teams for a proposed pedagogical-therapeutic intervention; xvi) harm reduction actions (030108031–3): Strategic actions aimed at minimizing harm caused by the use of different drugs without necessarily abstaining; xvii) follow-up of therapeutic residential service (030108032–1): Care supporting individuals residing in therapeutic residences, ensuring network articulation and promoting autonomy.

For summary purposes, consultations conducted for individuals with depression at CAPS were categorized into individual and group consultations based on the year of registration.

The number of individuals treated at CAPS was grouped according to their clinical and demographic profiles and stratified by year of registration in the RAAS. Clinically, this included the diagnosis of depression (depressive episodes and recurrent depressive disorders) and the type of care provided at CAPS (as described earlier). Demographically, gender (male and female), age group (18 to 40 years old; 41 to 60 years old; and $\geq$61 years old), and race/color (white, black, brown, yellow, indigenous, and unrecorded) were considered.

The rate of individuals diagnosed with depression treated at CAPS per 100,000 inhabitants was calculated as well, stratified by year (2013 and 2019) and federative unit. Population data were sourced from the Brazilian Institute of Geography and Statistics (IBGE) [23]. Additionally, the difference in the rate of individuals assisted in CAPS between 2019 and 2013 was calculated to assess which federative units experienced significant increases in this indicator.

The presumed coverage of CAPSs in the country was determined based on three parameters. Firstly, the number of active CAPSs registered in the National Register of Health Establishments (CNES/DATASUS). Secondly, the presumed coverage of CAPSs by type of care, as per parameters established by the Ministry of Health during the analyzed period [24, 25]. This includes CAPS I (50 thousand inhabitants), CAPS II (100 thousand inhabitants), CAPS III (150 thousand inhabitants), CAPS AD II (100 thousand inhabitants), CAPS AD III (150 thousand inhabitants), and for CAPS AD IV (500 thousand inhabitants), population size was used to qualify the establishment due to the absence of presumed coverage indication in the Ministry of Health standard. Thirdly, the population in the territory under investigation (Brazil and federative units). The number of CAPSs (parameter 1) was multiplied by the presumed coverage of care for each care modality (parameter 2), and then divided by the population of the territory of interest (parameter 3). This process was conducted for the years 2013 and 2019.

## Data analysis

The data collected in this study represent the number of individuals assisted in each investigated year, rather than the total number of consultations provided at CAPSs. Therefore, if an individual received multiple consultations within the same year, they were counted only once.

Absolute (quantity) and relative (percentage) frequencies of the analyzed variables were calculated and presented in tables and figures. The variation in the number of people assisted at CAPSs was analyzed by clinical profile (ICD-10 code and type of care) and demographic characteristics (gender, age group, race/color) over the period investigated (2013–2019). In this analysis, 2013 served as the base year, with all subsequent years calculated as percentages

relative to 2013. Thus, values less than one indicate a reduction in the number of individuals assisted at CAPSs compared to 2013, while values greater than one indicate an increase.

To illustrate the rate of individuals assisted at CAPSs and the presumed coverage, maps with state demarcations were generated to depict the geographic distribution. Excel® was utilized for this purpose, making use of map resources integrated within the software.

## Ethical aspects

As this study relies on secondary databases without individual participant identification, it does not fall under the purview of requiring approval from the Institutional Ethics Committee, as stipulated in Resolution No. 510/2016. Therefore, institutional ethics approval was not deemed necessary for this research.

## Results

The number of individuals receiving treatment for depression at CAPSs increased during the period under investigation, rising from 217,693 in 2013 to 451,789 in 2019, reflecting a growth rate of 107%. In absolute terms, there was a significant upsurge in the number of patients treated at CAPSs in 2019, indicating an increase of 101,335 individuals compared to the preceding year (Table 1).

In terms of clinical characteristics, most patients were diagnosed with depressive episodes (65%), followed by recurrent depressive disorders (35%) (Fig 1A). From 2013 to 2019, there was a slight increase in both diagnoses among patients treated at CAPSs. However, this increase was more significant for depressive episodes compared to recurrent depressive disorders (121% versus 85%, respectively) when comparing the number of patients in 2013 and 2019 (Fig 1B). Regarding the type of care provided at CAPSs, individualized care (75%) was more prevalent than collective care (25%). The collective procedure that exhibited the most notable growth during the analyzed period was family care, which increased by 245% between 2013 and 2019. Nonetheless, in absolute terms, the most frequently performed collective procedure was group care, provided for 57,412 patients in 2019 (Table 2).

**Table 1. Number of patients with a primary diagnosis of depression treated at CAPSs by demographic characteristics and year of care, Brazil, 2013–2019.**

| Patients assisted at CAPSs | 2013 | 2014 | 2015 | 2016 | 2017 | 2018 | 2019 |
|---|---|---|---|---|---|---|---|
| **Brazil** | 217,693 | 243,386 | 287,702 | 316,225 | 338,957 | 350,454 | 451,789 |
| **Sex** | | | | | | | |
| Female | 171,606 | 189,938 | 222,611 | 243,221 | 259,458 | 270,564 | 344,058 |
| Male | 46,087 | 53,448 | 65,091 | 73,004 | 79,499 | 79,890 | 107,731 |
| **Age range** | | | | | | | |
| 18–40 years old | 81,386 | 92,767 | 110,794 | 112,622 | 134,689 | 143,661 | 195,556 |
| 41–60 years old | 111,093 | 123,101 | 145,474 | 159,884 | 168,725 | 171,525 | 211,318 |
| ≥61 years old | 25,214 | 27,518 | 31,434 | 43,719 | 35,543 | 35,268 | 44,914 |
| **Race/color** | | | | | | | |
| White | 80,535 | 90,004 | 103,105 | 118,240 | 127,420 | 124,101 | 168,150 |
| Black | 8,514 | 9,447 | 10,509 | 11,553 | 12,970 | 13,566 | 18,303 |
| Brown | 65,926 | 76,700 | 94,053 | 95,214 | 106,620 | 119,665 | 149,052 |
| Yellow | 1,763 | 2,317 | 4,075 | 9,694 | 18,550 | 24,087 | 33,956 |
| Indigenous | 104 | 72 | 79 | 138 | 192 | 402 | 425 |
| No record | 60,851 | 64,846 | 75,881 | 81,386 | 73,205 | 68,633 | 81,903 |

**Source:** Outpatient Information System, through the Records of Outpatient Health Actions (SIA/RAAS).

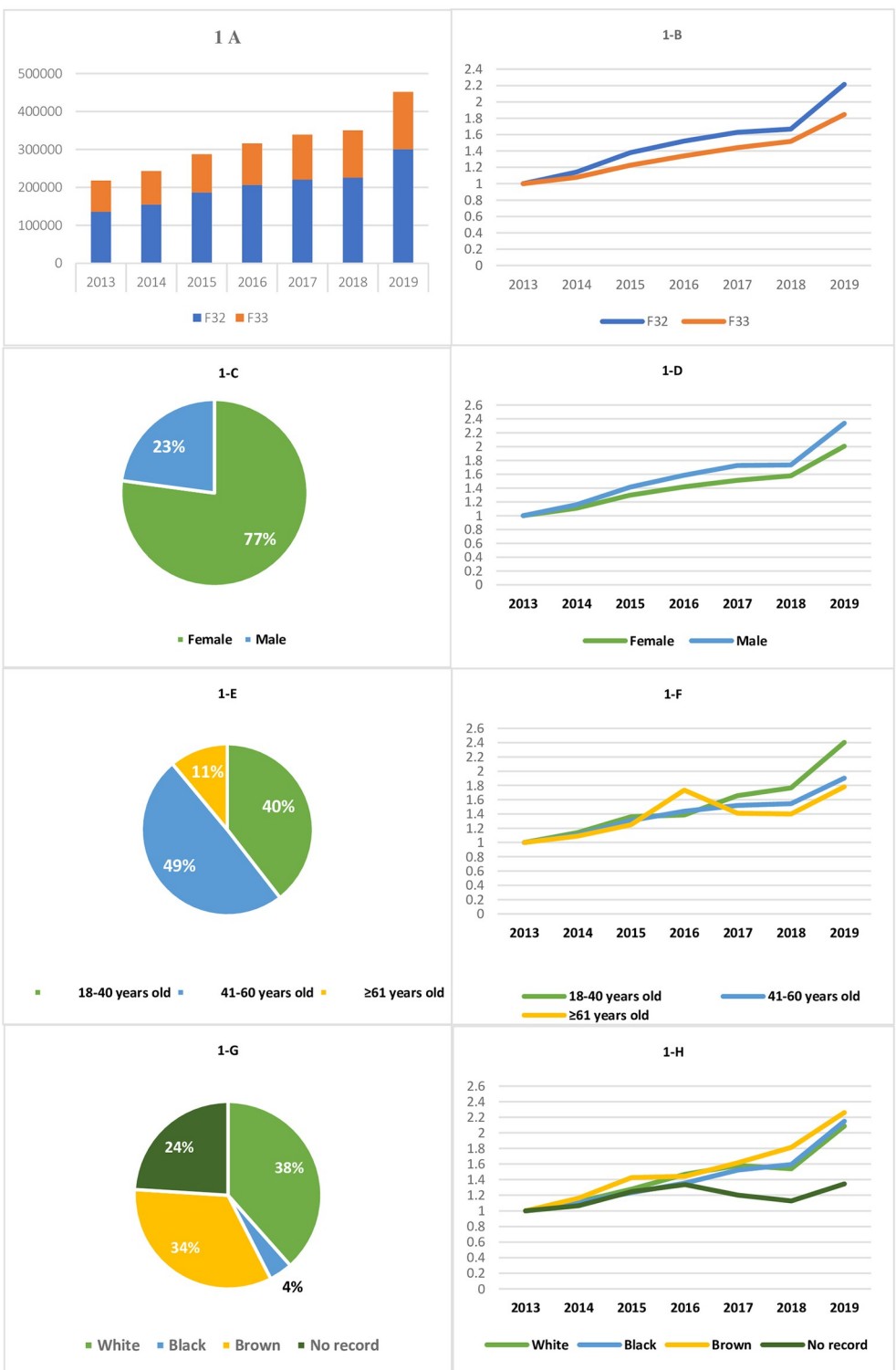

**Fig 1. Type of diagnosis of depression and profile of patients diagnosed with depression treated at CAPS, Brazil, 2013–2019. Source:** Outpatient Information System, through the Records of Outpatient Health Actions (SIA/RAAS).

**Table 2. Number of individuals with a primary diagnosis of depression treated at CAPSs by type and year of care, Brazil, 2013–2019.**

| Procedure code | Description | Quantity per year | | | | | | |
|---|---|---|---|---|---|---|---|---|
| | | 2013 | 2014 | 2015 | 2016 | 2017 | 2018 | 2019 |
| 30108002–0 | Night Reception | 71 | 94 | 127 | 134 | 209 | 220 | 383 |
| 30108003–8 | Reception in the third shift | 189 | 53 | 49 | 41 | 46 | 38 | 67 |
| 30108004–6 | Follow-up of a Patient in Residential Therapeutic Service | 110 | 80 | 88 | 123 | 109 | 101 | 222 |
| 30108019–4 | Daytime Reception of Patients in CAPS | 23,173 | 28,256 | 30,693 | 29,244 | 34,492 | 28,302 | 45,471 |
| 30108029–1 | Attention to Crisis Situations | 1,224 | 1,299 | 1,342 | 1,567 | 1,724 | 2,293 | 3,588 |
| 30108020–8 | Individual Patient Care at CAPS | 133,479 | 149,463 | 179,442 | 207,529 | 223,303 | 231,362 | 295,467 |
| | **Total patients (individual care)** | **158,246** | **179,245** | **211,741** | **238,638** | **259,883** | **262,316** | **345,198** |
| 30108021–6 | Patient Group Care at CAPS | 39,474 | 41,633 | 48,789 | 50,163 | 48,613 | 51,381 | 57,412 |
| 30108022–4 | Family Care in CAPS | 6,062 | 6,636 | 8,318 | 9,370 | 11,722 | 14,813 | 20,909 |
| 30108024–0 | Home visit | 2,073 | 2,186 | 2,790 | 3,156 | 3,345 | 3,475 | 4,106 |
| 30108027–5 | Body Practices | 3,769 | 3,949 | 4,101 | 4,401 | 4,577 | 5,392 | 6,041 |
| 30108028–3 | Expressive and Communicative Practices | 5,285 | 6,404 | 6,752 | 6,391 | 7,001 | 9,052 | 11,147 |
| 30108034–8 | Psychosocial Rehabilitation Actions | 2,194 | 2,173 | 2,551 | 3,029 | 2,451 | 2,607 | 4,635 |
| 30108035–6 | Promotion of Contractuality | 582 | 1,159 | 1,246 | 1,074 | 1,361 | 1,403 | 2,340 |
| | **Total patients (collective care)** | **59,439** | **64,140** | **74,547** | **77,584** | **79,070** | **88,123** | **106,590** |

**Source:** Outpatient Information System, through the Records of Outpatient Health Actions (SIA/RAAS).

Regarding the demographic profile, women comprised 77% of the patients treated at CAPSs throughout the analyzed period (Fig 1C), ranging from 76% in 2013 to 79% in 2019. The data indicate a slight trend towards a greater increase in male patients during this period (133.76% versus 100.49%, respectively) (Fig 1D). The age group of 41–60 years was the most prevalent, accounting for 49% of the patients attended to at CAPSs during the analyzed period (Fig 1E). Nevertheless, we observed more significant growth in the younger age group (18–40 years), particularly after 2017 (Fig 1F). Caucasian and brown individuals constituted 72% of the patients (Fig 1G), with brown individuals demonstrating more substantial growth through-out the analyzed period (Fig 1H).

The rate of individuals with a primary diagnosis of depression receiving assistance in CAPSs per 100,000 inhabitants was 109 in Brazil in 2013, with significant disparities observed between federative units, ranging from zero in Amapá to 235 in Santa Catarina (Table 3). This pattern of inequality further accentuated by 2019, with states exhibiting rates ranging from 1 (Amapá) to 466 patients per 100,000 inhabitants (Mato Grosso) (Table 3). Nationally, the rate increased to 215 per 100,000 population in 2019. Upon analyzing the difference in the rate of patients assisted per 100,000 inhabitants between 2013 and 2019, notable increases were observed in the states of Mato Grosso (323), Piauí (275), Rondônia (243), Mato Grosso do Sul (237), Pará (171), and Minas Gerais (153) (Table 3).

An increase in the presumed coverage of CAPSs across the country was noted, rising from 71% in 2013 to 87% in 2019. However, disparities persist among federative units in both years under investigation. In 2013, presumed coverage ranged from 31% (Acre) to 125% (Paraíba) (Table 4), while in 2019, it varied from 41% (Amazonas and Amapá) to 147% (Paraíba) (Table 4). In 2013, seven states had presumed coverage exceeding 90%, with five exceeding 100%. By 2019, 13 states achieved presumed coverage exceeding 90%, with nine surpassing 100%. It is worth noting that two states experienced a reduction in presumed coverage between 2013 and 2019 (Rondônia and Ceará), although the latter still maintained relatively high presumed coverage in 2019 (108%). The state exhibiting the most substantial growth in

**Table 3. Rate of patients with a primary diagnosis of depression treated at CAPS per 100,000 inhabitants, Brazil, 2013 and 2019.**

| State | 2013 | 2019 | Difference |
|:---:|:---:|:---:|:---:|
| AC | 16 | 74 | 58 |
| AL | 206 | 329 | 123 |
| AM | 46 | 120 | 74 |
| AP | 0 | 1 | 1 |
| BA | 109 | 212 | 103 |
| CE | 211 | 347 | 136 |
| DF | 8 | 51 | 43 |
| ES | 37 | 52 | 14 |
| GO | 89 | 198 | 110 |
| MA | 71 | 172 | 101 |
| MG | 111 | 264 | 153 |
| MS | 176 | 414 | 237 |
| MT | 142 | 466 | 324 |
| PA | 142 | 314 | 172 |
| PB | 180 | 278 | 98 |
| PE | 57 | 154 | 96 |
| PI | 103 | 378 | 276 |
| PR | 174 | 244 | 70 |
| RJ | 30 | 30 | 1 |
| RN | 83 | 131 | 48 |
| RO | 153 | 396 | 244 |
| RR | 15 | 127 | 112 |
| RS | 105 | 175 | 71 |
| SC | 235 | 357 | 122 |
| SE | 63 | 168 | 105 |
| SP | 103 | 216 | 113 |
| TO | 80 | 200 | 120 |

**Source:** Outpatient Information System (SIA/SUS), through Records of Outpatient Health Actions (SIA/RAAS) and Brazilian Institute of Geography and Statistics (IBGE).

presumed CAPS coverage was Minas Gerais, with an increase of 49 percentage points between 2013 and 2019 (Table 4).

## Discussion

Our study yielded several key findings. Firstly, we observed a significant increase of 107% in the number of individuals diagnosed with depression receiving treatment at CAPSs between 2013 and 2019. Secondly, there was an overall rise in the presumed coverage of CAPSs nationwide, climbing from 71% in 2013 to 87% in 2019, although notable disparities existed among federative units. Thirdly, we noted a more pronounced expansion in care provision for depressive episodes, indicating a focus primarily on crisis intervention when individuals present acute and exacerbated symptoms. Fourthly, individual consultations were more prevalent, reflecting a patient-centered approach to care. Notably, collective modalities of care, particularly those targeting family members, exhibited significant growth. Lastly, in terms of

**Table 4. Presumed coverage of Psychosocial Care Centers (CAPSs) by state according to attendances for depression, Brazil, 2013 and 2019.**

| State | 2013 | 2019 | Difference |
|---|---|---|---|
| | Presumed Coverage | Presumed Coverage | 2019–2013 |
| AC | 31% | 51% | 20% |
| AL | 99% | 112% | 14% |
| AM | 35% | 41% | 6% |
| AP | 34% | 41% | 8% |
| BA | 89% | 103% | 14% |
| CE | 115% | 108% | -8% |
| DF | 38% | 45% | 7% |
| ES | 45% | 52% | 7% |
| GO | 58% | 73% | 14% |
| MA | 66% | 77% | 11% |
| MG | 70% | 120% | 49% |
| MS | 58% | 88% | 30% |
| MT | 65% | 67% | 2% |
| PA | 56% | 62% | 6% |
| PB | 125% | 147% | 22% |
| PE | 66% | 90% | 24% |
| PI | 101% | 125% | 24% |
| PR | 67% | 94% | 27% |
| RJ | 52% | 65% | 13% |
| RN | 78% | 91% | 13% |
| RO | 75% | 62% | -14% |
| RR | 102% | 124% | 22% |
| RS | 94% | 122% | 28% |
| SC | 79% | 91% | 13% |
| SE | 108% | 128% | 20% |
| SP | 60% | 71% | 11% |
| TO | 68% | 89% | 21% |
| BRAZIL | 71% | 87% | 16% |

**Source:** Outpatient Information System (SIA/SUS), through Records of Outpatient Health Actions (SIA/RAAS) and Brazilian Institute of Geography and Statistics (IBGE).

demographic characteristics, we observed a predominance of assistance to women aged between 41 and 60 years old, and individuals of white and brown race/color.

Our findings revealed a notable disparity in the distribution of CAPSs across the national territory. Among the 27 federative units, 15 (55%) and 9 (33%) were below the coverage threshold considered as very good by the Ministry of Health in 2013 and 2019, respectively [25]. In 2019, the North region accounted for most of these states with presumed coverage below 70%, except for Tocantins (89%) and Roraima (124%). Previous studies have highlighted the challenges faced by the North region in implementing universal health care, primarily stemming from socioeconomic disparities and limited governance by health managers [26, 27].

We did not find any empirical studies supporting the parameters used by the Ministry of Health to define CAPS coverage by care modality. This discrepancy may not accurately reflect CAPS's actual capacity to serve the population and could lead to overly optimistic coverage estimations. In practice, the Ministry of Health utilizes different parameters to enable CAPS

based on more modest population thresholds [28]: CAPS I (≥20 thousand inhabitants); CAPS II (≥70 thousand inhabitants); CAPS III (≥200 thousand inhabitants); CAPS AD II (≥70 thousand inhabitants); CAPS AD III (≥150 thousand inhabitants); and CAPS AD IV (≥500 thousand inhabitants). For instance, while a CAPS I could be established in a municipality with at least 20 thousand inhabitants, the Ministry of Health arbitrarily defines the coverage of this facility for 50 thousand inhabitants. Hence, we qualify our findings as presumed coverage.

Care provided at CAPSs must adopt an interdisciplinary approach, encompassing individual consultations as well as collective interventions in groups and/or involving family members. Previous studies [29, 30] conducted from this perspective have underscored the importance of family involvement in treatment. Our results revealed an increase in consultations for family members, facilitating information exchange and guiding the treatment of users. This engagement serves as a pivotal axis for the care proposal offered at CAPSs, enabling individuals to reconstruct their lives beyond the realm of health and illness. This aspect contributes significantly to social rehabilitation and the reintegration of individuals into society.

Treatment for depression typically encompasses medication and group or individual therapies, which may be used in conjunction or separately based on the severity and duration of symptoms [2]. Our data indicated a notable increase in individual consultations, particularly for acute depression crises, alongside a rise in consultations for family members and group activities. The literature underscores that the selection of therapy depends on an assessment of symptom intensity and duration. However, previous studies have highlighted that less than half of individuals with depression receive appropriate treatment [31, 32].

The diagnosis of depression relies on clinical assessment, considering the subject's self-report encompassing emotional and physical aspects. However, it is imperative for professionals to remain vigilant for early detection and intervene appropriately [33]. These considerations may account for the substantial number of consultations observed for depressive episodes, wherein care is provided primarily during a crisis to alleviate main symptoms, albeit without ensuring complete remission of the condition. Several factors may impede continuity of care, including a limited number of services and adequately trained professionals, as well as challenges in sensitizing individuals to the importance of adhering to treatment for the agreed-upon duration rather than solely during crises [34, 35].

Depression disproportionately affects a higher proportion of women [36] compared to men, a trend supported by our study findings. However, we observed an increase in attendance among the male population. Studies conducted with a focus on men seeking health services for depression have revealed that traditional societal norms surrounding masculinity exert a significant influence. These norms can have a threefold greater impact on individuals experiencing depression, affecting the expression and management of symptoms, as well as attitudes, intentions, and actual help-seeking behavior [37, 38].

Regarding age, our findings indicate that approximately half of the consultations were for individuals aged 41–60. Notably, there was an increase in young people seeking treatment at CAPSs for depression after 2017. The involvement of young and economically active individuals with depression has surged, leading to higher rates of absenteeism [3, 39]. In terms of skin color, most users assisted were white, with an increase observed among those who self-identified as brown. This aligns with a previous study [36] evaluating depression prevalence in adults in Brazil, which found that young white individuals had the lowest depression prevalence. Furthermore, an evaluation of mental health service utilization by individuals with depression revealed that the majority (81%) were female, with a mean age of 49 years [40]. Thus, the typical characteristics of the subjects in our study closely resemble those reported in the literature.

Our study underscores the increasing attendance for depression treatment at CAPSs, which mirrors the global growth in the prevalence of this condition. Another possible explanation for

the significant increase in individuals seeking assistance at CAPSs may be the lack of preparedness among primary healthcare professionals to detect signs and symptoms when they are still mild and treatable at this level of care. Additionally, the limited number of healthcare facilities equipped to address such demands may contribute to this trend. In Brazil, healthcare aims to provide comprehensive assistance through interconnected healthcare networks, including the Psychosocial Care Network (RAPS). Mild and moderate depression cases should ideally be managed in primary healthcare settings, with only severe, treatment-resistant cases referred to specialized care, such as CAPSs. However, despite the Ministry of Health reporting that CAPS coverage per inhabitant is within acceptable standards, Brazil still faces gaps in care provision [41].

## Strengths and limitations of the study

Strengths of our study include its national scope, spanning seven years, which enabled us to discern advancements and challenges in managing patients affected by depression in CAPSs. Moreover, our findings are derived from real-world data, extracted from records maintained for patients treated at health services. Additionally, we calculated the presumed coverage of CAPS based on parameters established by the Ministry of Health, shedding light on the disparities among Brazilian federative units and the obstacles in achieving comprehensive coverage of the country's population.

However, our study has several limitations. Firstly, our results pertain solely to individuals assisted in CAPSs and do not encompass the entire mental health service network (RAPS) within the public health system. Secondly, our analysis is based on the number of individuals served rather than the number of appointments received, potentially underestimating the actual demand for services. Thirdly, there are opportunities for improvement in the accuracy of records within the outpatient health actions (RAAS-psychosocial) system, as data entry is still performed manually and may be susceptible to typographical errors [42]. Lastly, our data are confined to the public health system, thus excluding services rendered through supplementary health (private health insurance) or out-of-pocket arrangements.

## Final remarks and recommendations for health policy

Attendance for individuals with depression has increased throughout the analyzed period, underscoring the escalating prevalence of this disorder and its social impact. However, the heightened utilization of these community resources does not necessarily equate to improved accessibility or expanded coverage of assistance. This is exacerbated by recent shifts in mental health policy, wherein there are greater financial incentives for psychiatric hospitalizations, therapeutic communities, and closed services reminiscent of the asylum model previously practiced. The landscape of this field is rife with obstacles, including the pervasive stigma confronting those with mental health challenges, necessitating further discourse to implement public policies that cater to the widest possible segment of the Brazilian population. Given the vast territorial expanse of our country and the predominance of small municipalities, there exists a critical need for a coordinated network of services capable of addressing specific demands and nuances.

## Supporting information

**S1 Table. Calculation memory for presumed coverage of CAPS by states and Brazil in 2013.**
(DOCX)

**S2 Table. Calculation memory for presumed coverage of CAPS by states and Brazil in 2019.**
(DOCX)

## Author Contributions

**Conceptualization:** Bruna Paiva do Carmo Mercedes, Everton Nunes da Silva, Rodrigo Luiz Carregaro, Adriana Inocenti Miasso.

**Data curation:** Bruna Paiva do Carmo Mercedes, Everton Nunes da Silva, Rodrigo Luiz Carregaro, Adriana Inocenti Miasso.

**Formal analysis:** Bruna Paiva do Carmo Mercedes, Everton Nunes da Silva, Adriana Inocenti Miasso.

**Investigation:** Bruna Paiva do Carmo Mercedes, Everton Nunes da Silva, Rodrigo Luiz Carregaro, Adriana Inocenti Miasso.

**Methodology:** Bruna Paiva do Carmo Mercedes, Everton Nunes da Silva, Rodrigo Luiz Carregaro, Adriana Inocenti Miasso.

**Project administration:** Bruna Paiva do Carmo Mercedes, Everton Nunes da Silva.

**Resources:** Bruna Paiva do Carmo Mercedes, Everton Nunes da Silva.

**Software:** Bruna Paiva do Carmo Mercedes, Everton Nunes da Silva.

**Supervision:** Bruna Paiva do Carmo Mercedes, Everton Nunes da Silva, Rodrigo Luiz Carregaro, Adriana Inocenti Miasso.

**Validation:** Bruna Paiva do Carmo Mercedes, Everton Nunes da Silva, Rodrigo Luiz Carregaro, Adriana Inocenti Miasso.

**Visualization:** Bruna Paiva do Carmo Mercedes, Everton Nunes da Silva.

**Writing – original draft:** Bruna Paiva do Carmo Mercedes, Everton Nunes da Silva.

**Writing – review & editing:** Bruna Paiva do Carmo Mercedes, Everton Nunes da Silva, Rodrigo Luiz Carregaro, Adriana Inocenti Miasso.

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
