## [Decision Letter · Decision Letter 0]

11 Mar 2024

PONE-D-24-02972Psychosocial Care Centers (CAPS) in Brazil: profile of individuals served and presumed coverage in the period 2013-2019PLOS ONE

Dear Dr. Paiva do Carmo Mercedes,

Thank you for submitting your manuscript to PLOS ONE. After careful consideration, we feel that it has merit but does not fully meet PLOS ONE’s publication criteria as it currently stands. Therefore, we invite you to submit a revised version of the manuscript that addresses the points raised during the review process.

the manuscript is very interesting, it meets most of the criteria, however it is suggested:

1. In the introduction add more references that allow you to give solid arguments to the topic of the article.

2. In the methodology, make the inclusion/exclusion criteria clear

3. Improve the discussion, highlighting key findings, adding references and clearly describing limitations.

4. In general, consider reviewing the manuscript for grammar and sentence coherence.

We look forward to receiving your revised manuscript.

Kind regards,

Oriana Rivera-Lozada de Bonilla

Academic Editor

PLOS ONE

4. We note that Figures 2 and 3 in your submission contain [map/satellite] images which may be copyrighted. All PLOS content is published under the Creative Commons Attribution License (CC BY 4.0), which means that the manuscript, images, and Supporting Information files will be freely available online, and any third party is permitted to access, download, copy, distribute, and use these materials in any way, even commercially, with proper attribution. For these reasons, we cannot publish previously copyrighted maps or satellite images created using proprietary data, such as Google software (Google Maps, Street View, and Earth). For more information, see our copyright guidelines: http://journals.plos.org/plosone/s/licenses-and-copyright.

a. You may seek permission from the original copyright holder of Figures 2 and 3 to publish the content specifically under the CC BY 4.0 license. 

Reviewers' comments:

Reviewer's Responses to Questions

**Comments to the Author**

1. Is the manuscript technically sound, and do the data support the conclusions?

Reviewer #1: Yes

Reviewer #2: Yes

2. Has the statistical analysis been performed appropriately and rigorously? 

Reviewer #1: Yes

Reviewer #2: N/A

3. Have the authors made all data underlying the findings in their manuscript fully available?

Reviewer #1: Yes

Reviewer #2: Yes

4. Is the manuscript presented in an intelligible fashion and written in standard English?

Reviewer #1: Yes

Reviewer #2: No

5. Review Comments to the Author

Reviewer #1: The manuscript covers a very interesting topic in context to community medicine and social care. The following minor recommendations are suggest:

Introduction:

Add a few references to the highlight the global perspective on the theme of the study

Discussion: The key findings and the limitations of the study should be clearly described.

Reviewer #2: The study set out to describe various characteristics of patients that attended CAPSs and calculate the presumed coverage of CAPSs. It was a descriptive study that took into account national data. I’ve put my comments based on the sections.

In the introduction section:

In the abstract we’re using “assumed coverage” but in the rest of the manuscript it’s “presumed coverage”. Use similar terms as it can get confusing.

Consider rewriting the objectives to better reflect the descriptive nature of the study.(For example say “to describe the profile…” for the first objective)

Please cite “Across the Americas, depression tops as one of the most disabling illnesses, accounting for 7.8% of total disability in the population.”

“In Brazil, in a national survey carried out in 2019, a prevalence of 10.2% of cases of depression was found in people aged 18 years or over “ what was the initial number? Use numbers to show the growth.

“There has been an expansion of CAPS in the country. In 2006, Brazil had 739

CAPSs, increasing to 2.209 in 2014 and 2.306 in 2018” are these decimal points or typos?

In the methodology section:

Have clear exclusion/inclusion criteria

Regarding study population: was there a reason for using 18years as a cut-off point?

Give more clarification for why the data after 2019 wasn’t considered representative.

For patients with recurrent depression are you considering those with previous visits or those with prior episoides but no prior visits?

In results section:

“ In 2013, 7 states had presumed coverage greater than 90%, with 5 having coverage greater than 100%. In 2019, the number of states was 13 and 9, respectively.” what do the number 13 and 9 represent?

In the discussion section:

Are claiming that number of visits increased because of increase in burden of depression or lack of mental health care service in the PHC units?

“The involvement of young and economically active people with depression has increased, resulting in higher absenteeism 13, 37.” Shouldn't this show decreasing trend of recurrent depression? Or be reflected in decreasing trends of repeat visits? which wasn’t considered in this study

Have a header for the conclusion and recommendations section.

If it’s possible and available can you use translated versions for the references ?

Overall consider revising the grammar and sentences coherence of the manuscript.

6. PLOS authors have the option to publish the peer review history of their article (what does this mean?). If published, this will include your full peer review and any attached files.

Reviewer #1: No

Reviewer #2: No

---

## [Author Response · Author response to Decision Letter 0]

26 Jun 2024

Questions Reviewer 1 Reviewer 2

Is the manuscript technically sound, and do the data support the conclusions? Yes Yes

Has the statistical analysis been performed appropriately and rigorously? Yes N/A

Have the authors made all data underlying the findings in their manuscript fully available? Yes Yes

Is the manuscript presented in an intelligible fashion and written in standard English? Yes No

Do you want your identity to be public for this peer review? No No

 Comments from Editor

Editor, comment 1: “Thank you for submitting your manuscript to PLOS ONE. After careful consideration, we feel that it has merit but does not fully meet PLOS ONE’s publication criteria as it currently stands. Therefore, we invite you to submit a revised version of the manuscript that addresses the points raised during the review process. the manuscript is very interesting, it meets most of the criteria, however it is suggested:1. In the introduction add more references that allow you to give solid arguments to the topic of the article. 2. In the methodology, make the inclusion/exclusion criteria clear. 3. Improve the discussion, highlighting key findings, adding references and clearly describing limitations. 4. In general, consider reviewing the manuscript for grammar and sentence coherence.”

Our response: Thank you for the opportunity to clarify the points raised by the reviewers. We provided a point-by-point reply to reviewers’ comments as reported below. We also complied with all journal requirements related to, ethical clearance (use of publicly available data), illustrations (all tables and figures were created by the authors of manuscript). Based on that, no copyright is applicable. Regarding the data availability, we included the calculation memory in the Supplementary Material.

Comments from reviewer 1

Reviewer 1, comment 1: “The manuscript covers a very interesting topic in context to community medicine and social care.”

Our response: We would like to thank you for your consideration and time spent reviewing our manuscript.

Reviewer 1, comment 2: “The following minor recommendations are suggest: Introduction: Add a few references to the highlight the global perspective on the theme of the study”

Our response: Thank you for your comment. We included more information on the burden of depression globally, as suggested by the reviewer. 

“Currently, depression is the leading mood disorder present in society. This condition affects more than 280 million people worldwide, and it is estimated that around 5% of adults suffer from depression 1. The World Health Organization (WHO) points to depression as one of the significant causes of disability and that it generates more impact on the global burden of disease 2, leading to high costs for the health and social security systems, as well as early mortality due to suicide 3. Depressive disorders primarily affect the young and economically active population 4. Across the Americas, depression tops as one of the most disabling illnesses, accounting for 7.8% of total disability in the population 3. Concerning South America, the ranking of disability from depression consists of five countries: Paraguay (9.4%), Brazil (9.3%), Peru (8.6%), Ecuador (8.3%) and Colombia (8.2%) 4.”

Reviewer 1, comment 3: “Discussion: The key findings and the limitations of the study should be clearly described.”

Our response: Thank you for your comment. We included a header in the Discussion to highlight the strengths and limitations of the study. We also added more clarifications on the limitations, as suggested by the reviewer. Regarding the key-findings, we summarized them in the first paragraph of the discussion. We rephased this paragraph to make it clear to readers, as recommend by the reviewer, which we transcribe below.

“There are several key-findings of our study. First, this study showed that the number of individuals with a primary diagnosis of depression treated at CAPSs increased by 107% between 2013 and 2019. Second, there was also an augmentation in the presumed coverage of CAPSs in the national territory, increasing from 71% in 2013 to 87% in 2019, with marked disparities between the federative units. Third, we found a more substantial expansion in care for depressive episodes, which suggests care based mainly on the crisis when the subjects present acute conditions and exacerbated signs and symptoms. Fourth, individual consultations were more frequent, configuring subject-centered assistance. For assistance in the collective modality, those aimed at family members stood out. Fifth, in demographic terms, assistance to women aged between 41 and 60 and of white and brown color/race prevailed.”

“Strengths and limitations of the study”

“As strengths, our study had a national scope and covered seven years, which allowed for identifying advances and challenges in managing patients affected by depression in CAPSs. In addition, the results come from real-world data, from records made for patients treated at health services. Additionally, our study calculated the presumed coverage of CAPS based on parameters established by the Ministry of Health, which allowed showing the inequalities between Brazilian federative units and the challenges to achieve full coverage of the country's population.”

“Regarding limitations, our results are restricted to individuals assisted in CAPSs and not to the entire service network for mental health (RAPS) in the public health system. Second, we base our results on the number of individuals served rather than the number of appointments received. Third, there are areas for improvement in the records of outpatient health actions (RAAS-psychosocial), as they are still carried out manually and subsequently transferred to the computerized system, which may lead to typing errors 40. Fourth, our data are also restricted to public health system, thus excluding services performed in supplementary health (private health insurance) or out-of-pocket.”

Comments from reviewer 2

Reviewer 2, comment 1: “The study set out to describe various characteristics of patients that attended CAPSs and calculate the presumed coverage of CAPSs. It was a descriptive study that took into account national data. I’ve put my comments based on the sections.”

Our response: We would like to thank you for your consideration and time spent reviewing our manuscript.

Reviewer 2, comment 2: “In the introduction section: In the abstract we’re using “assumed coverage” but in the rest of the manuscript it’s “presumed coverage”. Use similar terms as it can get confusing.”

Our response: Thank you for your comment. We replaced “assumed” by “presumed” as suggested by the reviewer. 

Reviewer 2, comment 3: “Consider rewriting the objectives to better reflect the descriptive nature of the study. (For example say “to describe the profile…” for the first objective)”.

Our response: Thank you for your comment. We replaced “analyze” by “describe” as suggested by the reviewer in the abstract and introduction sections. 

Reviewer 2, comment 4: “Please cite “Across the Americas, depression tops as one of the most disabling illnesses, accounting for 7.8% of total disability in the population.””.

Our response: Thank you for your comment. We included the citation for the information mentioned by the reviewer. 

Reviewer 2, comment 5: ““In Brazil, in a national survey carried out in 2019, a prevalence of 10.2% of cases of depression was found in people aged 18 years or over “what was the initial number? Use numbers to show the growth.”

Our response: Thank for your comment. We included the information as suggested by the reviewer. 

“In Brazil, the prevalence of depression was 7.6% for individuals aged 18 years or over in 2013, reached 10.2% in 2019, which represented an increase of 34.2% in 6-year period.”

Reviewer 2, comment 6: ““There has been an expansion of CAPS in the country. In 2006, Brazil had 739 CAPSs, increasing to 2.209 in 2014 and 2.306 in 2018” are these decimal points or typos?”

Our response: Sorry for these typos. The correct figures are “increasing to 2,209 in 2014 and 2,306 in 2018”. 

Reviewer 2, comment 7: “In the methodology section: Have clear exclusion/inclusion criteria”.

Our response: Thank you for your comment. We made it clearer as suggested by the reviewer. 

“The study population was based on the following inclusion criteria: 

i) individuals with a medical diagnosis of depression according to the ICD-10, with codes F32-F32.9 (depressive episodes) and F33-F33.9 (recurrent depressive disorders). It was considered only ICD-10 codes based on the principal diagnosis, which occasioned the need for the outpatient care; 

ii) individuals aged 18 years or older. We opted for this cut-off because CAPSs are essentially design to adult population. Few CAPSs are targeted to provide mental care to children and adolescents, focusing on their suffering and cognitive development.

iii) individuals that used the CAPSs in any moment from 2013 to 2019. 

The exclusion criteria are: 

i) individuals that received care in CAPSs but by any reason they were not registered in the administrative records from the Ministry of Health. Some reasons may be the fail to provide all information required by the Ministry of Health such as ICD-10 code, ID of the patient. 

ii) Individuals that have used any other service than the one provided by the CAPSs.”

Reviewer 2, comment 8: “Regarding study population: was there a reason for using 18years as a cut-off point?”

Our response: Thank you for your comment. We added the reason why we opted for 18 years as a cut-off, as described below. 

“We opted for this cut-off because CAPSs are essentially design to adult population. Few CAPSs are targeted to provide mental care to children and adolescents, focusing on their suffering and cognitive development.”

Reviewer 2, comment 9: “Give more clarification for why the data after 2019 wasn’t considered representative.”

Our response: Thank you for your comment. We added more clarification on the point raised by the reviewer, as transcribed below. 

“The choice of the period was based on the year in which the consultations carried out in the CAPS started to be registered in this application (2013) and the last year before the COVID-19 pandemic (2019). The data entered the system from 2020 onwards could not represent the reality of care, given the change in the functioning of health devices caused by the pandemic. During the pandemic period, health authorities have implemented several guidelines aiming at tackling the pandemic, including reallocation of healthcare workers and supplies to the COVID-19 effort, which have reduced the delivery of services not related to COVID-19. Moreover, population had concerns about pandemic, causing delay or avoidance of medical care as a way to reduce the risk of be infected by SARS-CoV-2.” 

Reviewer 2, comment 10: “For patients with recurrent depression are you considering those with previous visits or those with prior episoides but no prior visits?”

Our response: Thank for the opportunity to clarify this point. We used the classification provided in the administrative records, which are based on the ICD-10 definition: “A disorder characterized by repeated episodes of depression as described for depressive episode (F32.-), without any history of independent episodes of mood elevation and increased energy (mania). There may, however, be brief episodes of mild mood elevation and overactivity (hypomania) immediately after a depressive episode, sometimes precipitated by antidepressant treatment. The more severe forms of recurrent depressive disorder (F33.2 and F33.3) have much in common with earlier concepts such as manic-depressive depression, melancholia, vital depression and endogenous depression. The first episode may occur at any age from childhood to old age, the onset may be either acute or insidious, and the duration varies from a few weeks to many months.”

Reviewer 2, comment 11: “In results section: “ In 2013, 7 states had presumed coverage greater than 90%, with 5 having coverage greater than 100%. In 2019, the number of states was 13 and 9, respectively.” what do the number 13 and 9 represent?”

Our response: Thank you for the opportunity to clarify this information. 13 refers to the number of states that reached presumed coverage greater than 90%, of which 9 reached presumed coverage equal or greater than 100%. We edited this sentence to make it more understandable. 

Reviewer 2, comment 12: “In the discussion section: Are claiming that number of visits increased because of increase in burden of depression or lack of mental health care service in the PHC units?”

Our response: Thank you for the opportunity to clarify this issue. In fact, both situations are acting as drivers to increase the delivery of mental care from CAPSs. We edited the paragraph to make it clearer. 

“Our study points out that the attendance for depression in the CAPS has increased, and this finding is in line with the growth in the prevalence of this pathology worldwide. Another possible justification for the significant increase in the number of individuals assisted in CAPSs would be the lack of preparation of professionals working in PHC to detect signs and symptoms when they are still mild and likely to be treated at this level of care, in addition to the reduced number of health facilities that meet such demands. Health care in Brazil aims at integral assistance through articulated healthcare networks. To meet mental health demands, there is the Psychosocial Care Network (RAPS). Cases of mild and moderate depression should be resolved in the PHC, and only severe cases that are resistant to treatment are referred to specialized care, in this case, the CAPS. Although the Ministry of Health reports that the CAPS/inhabitant coverage rate is within the standards considered adequate, Brazil has gaps in care 39.”

Reviewer 2, comment 13: ““The involvement of young and economically active people with depression has increased, resulting in higher absenteeism 13, 37.” Shouldn't this show decreasing trend of recurrent depression? Or be reflected in decreasing trends of repeat visits? which wasn’t considered in this study”

Our response: Thank for raised this issue. Both interpretations may be possible, but the descriptive design of the study do not allow us to conclude it, which is out of the scope of the manuscript. However, these questions are interesting points for future research. 

Reviewer 2, comment 14: “Have a header for the conclusion and recommendations section.”

Our response: Thank you for your comment. We included a header from final remarks and recommendations for health police, which we transcribe below. 

“Final remarks and recommendation to health policy”

“Attendances for subjects with depression increased throughout the analyzed period, reinforcing the greater involvement of this disorder and its social burden; however, the greater use of these community devices does not represent greater availability of access or assistance coverage, even more so with the recent changes in the current mental health policy, with greater financial incentives for psychiatric hospitalizations and therapeutic communities, closed services and similar to the asylum model of care previously carried out. This field is permeated by obstacles, even the stigma faced by individuals with mental suffering, requiring further discussions to implement public policies that meet the greatest possible number of the Brazilian population. We have a country with a large territorial extension and with most of the municipalities being small, hence the importance of an articulated network of services that meet the demands according to the particularities presented.”

Reviewer 2, comment 15: “If it’s possible and available can you use translated versions for the references?”

Our response: Thank you for your comment. We included the English

---

## [Decision Letter · Decision Letter 1]

22 Jul 2024

Profile of Individuals Served and Presumed Coverage of Psychosocial Care Centers (CAPS) in Brazil: A Study of the Period 2013-2019.

PONE-D-24-02972R1

Dear Dr.Bruna Paiva do Carmo Mercedes ,

We’re pleased to inform you that your manuscript has been judged scientifically suitable for publication and will be formally accepted for publication once it meets all outstanding technical requirements.

Kind regards,

Oriana Rivera-Lozada de Bonilla

Academic Editor

PLOS ONE

**Comments to the Author**

1. If the authors have adequately addressed your comments raised in a previous round of review and you feel that this manuscript is now acceptable for publication, you may indicate that here to bypass the “Comments to the Author” section, enter your conflict of interest statement in the “Confidential to Editor” section, and submit your "Accept" recommendation.

Reviewer #1: All comments have been addressed

2. Is the manuscript technically sound, and do the data support the conclusions?

Reviewer #1: Yes

3. Has the statistical analysis been performed appropriately and rigorously? 

Reviewer #1: Yes

4. Have the authors made all data underlying the findings in their manuscript fully available?

Reviewer #1: Yes

5. Is the manuscript presented in an intelligible fashion and written in standard English?

Reviewer #1: Yes

6. Review Comments to the Author

Reviewer #1: Thank you for addressing the comments. The manuscript is now suitable for publication in the journal.

7. PLOS authors have the option to publish the peer review history of their article (what does this mean?). If published, this will include your full peer review and any attached files.

Reviewer #1: No

---

## [Editor Report · Acceptance letter]

25 Jul 2024

PONE-D-24-02972R1 

PLOS ONE

Dear Dr. Paiva do Carmo Mercedes, 

I'm pleased to inform you that your manuscript has been deemed suitable for publication in PLOS ONE. Congratulations! Your manuscript is now being handed over to our production team.

Kind regards, 

on behalf of

Dr. Oriana Rivera-Lozada de Bonilla 

Academic Editor

PLOS ONE